# Significance of skull osteoporosis to the development of peritumoral brain edema after LINAC-based radiation treatment in patients with intracranial meningioma

**Ryang-Hun Lee[1], Jae Min Kim[1], Jin Hwan Cheong[1], Je Il Ryu[1], Young Soo Kim[2], Myung-Hoon Han[1]***

**1** Department of Neurosurgery, Hanyang University Guri Hospital, Guri, Gyonggi-do, Korea, **2** Department of Neurosurgery, Hanyang University Medical Center, Seongdong-gu, Seoul, Korea

\* gksmh80@gmail.com

## Abstract

### Background and purpose

Disruption of the tumor-brain barrier in meningioma plays a critical role in the development of peritumoral brain edema (PTBE). We hypothesized that osteoporotic conditions may be associated with PTBE occurrence after radiation in patients with intracranial meningioma.

### Methods

We measured Hounsfield units (HU) of the frontal skull on simulation brain CT in patients who underwent linear accelerator (LINAC)-based radiation treatment for intracranial meningioma. Receiver operating characteristic curve analysis was performed to determine the optimal cut-off values for several predictive factors. The cumulative hazard for PTBE was estimated and classified according to these factors. Hazard ratios were then estimated to identify independent predictive factors associated with the development of PTBE after radiation in intracranial meningioma patients.

### Results

A total of 83 intracranial meningiomas in 76 patients who received LINAC-based radiation treatment in our hospital over an approximate 5-year period were included for the study. We found mean frontal skull HU $\leq$630.625 and gross tumor volume >7.194 cc to be independent predictors of PTBE after radiation treatment in patients with meningioma (hazard ratio, 8.41; $P$ = 0.019; hazard ratio, 5.92; $P$ = 0.032, respectively). In addition, patients who were $\geq$65 years showed a marginally significant association with PTBE.

### Conclusions

Our study suggests that possible osteoporotic conditions, large tumor volume, and older age may be associated with PTBE occurrence after LINAC-based radiation treatment for intracranial meningioma. In the future we anticipate that these findings may enhance the

**Data Availability Statement:** All relevant data are within the manuscript and its Supporting Information files.

**Funding:** This study was funded by Hanyang University (KR) (HY- 201900000003370). The funders had no role in study design, data collection and analysis, decision to publish, or preparation of the manuscript.

**Competing interests:** The authors have declared that no competing interests exist.

understanding of the underlying mechanisms of PTBE after radiation in meningioma patients.

## Introduction

Meningiomas are the most common extra-axial primary intracranial benign tumors and account for 13–26% of all primary intracranial tumors [1]. Although microsurgical tumor resection is the treatment of choice for symptomatic meningiomas, gross total resection of meningiomas is not always possible due to various conditions such as tumor size, location, adjacent neurovascular structures, or the patient's medical status. Radiation therapy is used as a treatment for meningiomas when the remnant tumor is present after surgery or when surgical resection is not an option [2]. Radiotherapy for meningioma is accepted as a safe treatment modality. Approximately 5% to 40% of patients experience treatment-related complications [3]. It was reported that symptomatic brain edema occurs in 37.5% of patients with parasagittal meningiomas after gamma knife radiosurgery [4]. Previously, several risk factors associated with peritumoral brain edema (PTBE) after radiosurgery in meningioma were reported. These include greater radiation dose, greater tumor size or volume, tumor location, brain-tumor interface, no prior resection for meningioma, and presence of pretreatment edema [5].

Disruption of the tumor-brain barrier in meningioma plays a critical role in the development of PTBE [6]. A previous study regarding microscopic anatomy of the brain–meningioma interface reported the presence of arachnoid trabeculae at the brain–meningioma contact interface [7]. We previously demonstrated a close correlation between bone mineral density (BMD) and Hounsfield unit (HU) values [8]. In addition, we suggested that systemic osteoporosis is negatively associated with the integrity of arachnoid trabeculae as both the bone and the arachnoid trabeculae are composed of type 1 collagen [8,9]. We hypothesized that osteoporotic conditions may be associated with PTBE after radiation in intracranial meningioma patients. To the best of our knowledge, there are no previous studies describing the possible relationship between osteoporotic conditions and PTBE after radiotherapy in meningioma which have been published [5].

To test this hypothesis, we measured HU values in the frontal bone from simulation brain computed tomography (CT) of patients who underwent linear accelerator (LINAC)-based radiation treatment for intracranial meningioma in our hospital. We evaluated other predictive risk factors for PTBE in meningioma after radiation treatment.

## Methods

### Study patients

This study was approved by the Institutional Review Board of Hanyang University Medical Center, Korea, and conformed to the tenets of the Declaration of Helsinki. Owing to the retrospective nature of the study, the need for informed consent was waived. All patient records were anonymized prior to analysis.

We retrospectively extracted data for all consecutive patients who were diagnosed with intracranial meningioma and received LINAC-based radiation treatment for the first time from the database of our hospital's NOVALIS registry, from July 7, 2014 to July 31, 2019. The registry has been designed for prospective research since July 7, 2014. Demographic patient information, prescribed radiation dose, and fractionation data were extracted from the NOVALIS registry.

All intracranial meningiomas were diagnosed by radiologic findings or histological confirmation following resection. All radiologic findings were confirmed by experienced neuro-radiologists. We only included patients with meningioma who underwent at least one follow-up imaging (CT/magnetic resonance imaging [MRI]) after LINAC-based radiation treatment in order to assess the occurrence of PTBE. The last imaging follow-up period after treatment was investigated in all study patients. PTBE was defined as the radiological confirmation of newly developed PTBE or the progression of preexisting PTBE after radiation treatment with newly developed neurological deficits. All patients had no preexisting PTBE among the patients who did not underwent surgery for meningioma before radiation treatment. Two patients were excluded due to no measurable cancellous bone of the frontal skull on brain CT.

## Radiation technique

All patients were treated using the NOVALIS Tx system (Varian Medical Systems, CA, USA; Brainlab, Feldkirchen, Germany) in our hospital. A noninvasive thermoplastic mask was used to perform simulation-computed tomography (CT) for radiation treatment. The Novalis Exac-Trac image system and robotic couch of the NOVALIS Tx system allowed us to adjust the patients' positions according to the information from the real-time image acquisition. Patients were treated with a 6 MV LINAC-based radiation treatment within 1 week from the day when the CT simulation was performed.

Gross tumor volume (GTV) was defined as the contrast-enhanced area on T1-weighted MRI images. In surgery patients, the GTV was defined as the postoperative resection cavity and the area of residual tumor in cases of subtotal resection. The clinical target volume (CTV) was identical to the GTV. The planning target volume (PTV) was defined as a symmetrical 0 to 2-mm expansion from the CTV. In case the tumor was located near an organ at risk, we adjusted the PTV with no expansion in the area of the tumor that was close to the organ at risk. The iPlan (Brainlab, Feldkirchen, Germany) and Eclipse (Varian, CA, USA) that are 3D treatment/planning systems of the NOVALIS Tx, were used for radiation planning using MRI/CT-fusion images in all intracranial meningioma patients. The 3D treatment/planning system automatically calculated the GTV, CTV, and PTV in all treated patients. We attempted to achieve tight conformality of the treatment isodose to the 3D reconstructed meningioma geometry.

Stereotactic radiosurgery (SRS) was defined as a single session treatment, hypofractionated SRS (hf-SRS) as 2 to 5 fractions, hypofractionated stereotactic radiotherapy (hFSRT) as 6 to 10 fractions, and fractionated stereotactic radiotherapy (FSRT) as doses delivered in >10 sessions (1.8–2.0 Gy/fraction) [10,11]. The biologically equivalent dose (BED) for the tumor was calculated according to the following equation: BED = $nd \times (1 + d/3)$, where n is the number of fractions, d is the dose per fraction, and $\alpha/\beta = 3$ [12].

## Measurement of frontal skull HU

Simulation-CT images (Philips Brilliance Big Bore CT Simulators) for radiation planning were used to measure the frontal skull HU values in all study patients. A previous study reported that variations in HU values across five CT scanners were in the range of 0–20 HU [13]. We previously demonstrated detailed methods for measuring HU values at each of four lines on the frontal cancellous bone. This was between the right and left coronal sutures on axial CT slices at the point where the lateral ventricles disappear [8,14]. The HU value of the frontal cancellous bone was measured using the "Linear histogram graph" function in the picture archiving and communication system (PACS) (PiViewSTAR version 5.0, INFINITT Healthcare, Seoul, Korea). The PACS automatically calculated the maximum, minimum, and mean HU

values according to the values on the drawn line. We recorded the mean HU value on each line of cancellous bone at the frontal bone region (Fig 1).

To avoid including cortical bone, all brain CT images were magnified for HU measurement. All frontal skull HU measurements were conducted by a trained neurosurgeon blinded to the clinical data of all patients.

## Other study variables

Clinical data including height, weight, hypertension, and diabetes were extracted from electronic medical records. Body mass index (BMI) was calculated as weight/ (height × height) and expressed in $kg/m^2$. Tumor location was confirmed by neuro-radiologists using the PACS.

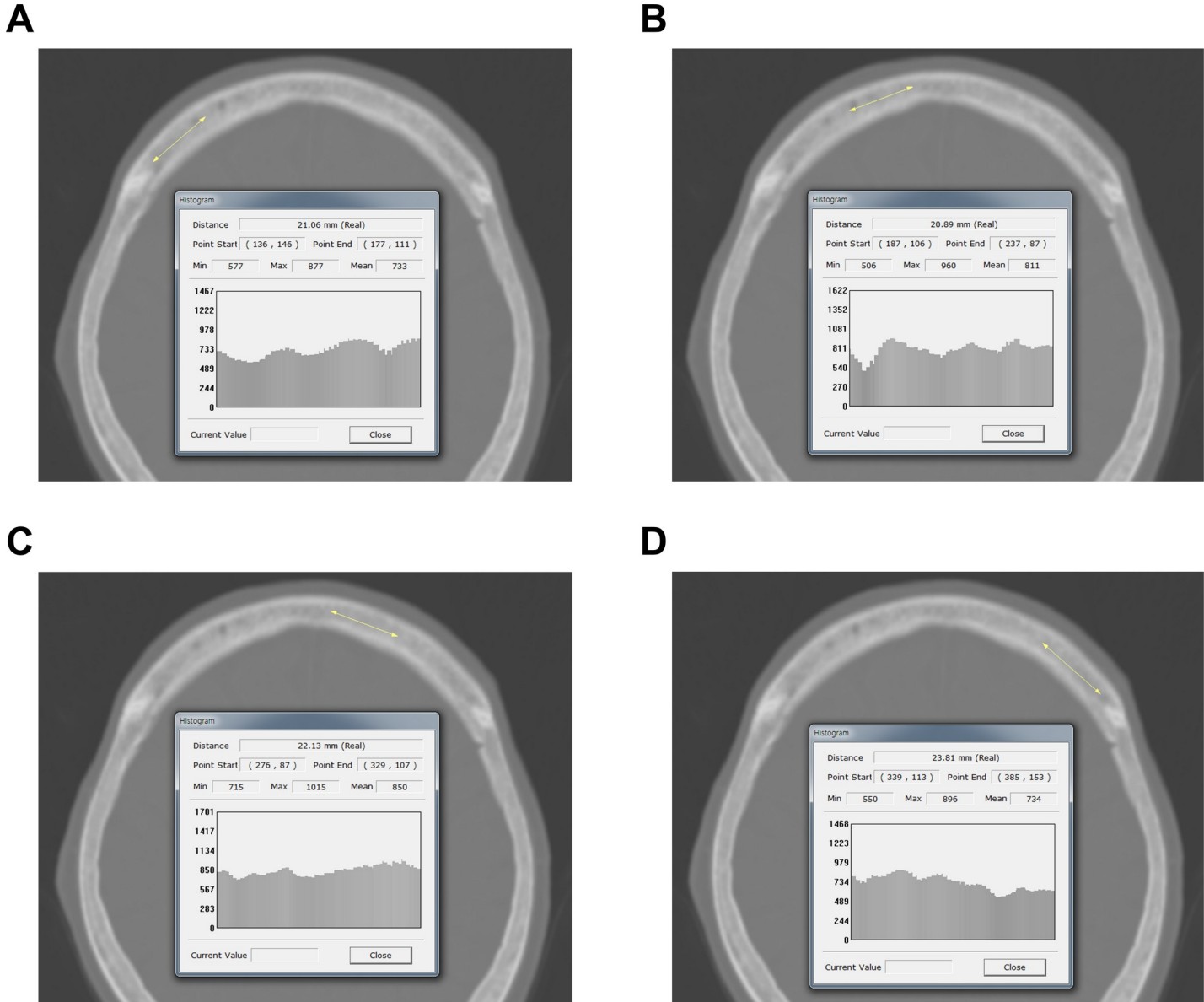

**Fig 1. Measurement of HU values at each of four lines on the frontal bone.** The PACS automatically calculated the maximum, minimum, and mean HU values according to the values on the drawn line. The mean HU value on each of the four lines was recorded. (A) Right lateral; (B) right medial; (C) left medial; (D) left lateral. HU = Hounsfield unit; PACS = picture archiving and communication system.

## Statistical methods

Continuous variables were expressed as mean ± SD or median with an interquartile range (IQR) and categorical variables were expressed as counts and percentage. The chi-square test and Student's t-test were used to assess statistical differences between non-PTBE and PTBE groups. The mean frontal skull HU value ([mean right lateral HU + mean right medial HU + mean left medial HU + mean left lateral HU]/4) was used in all analyses.

Receiver operating characteristic (ROC) curve analysis was performed to determine the optimal cut-off values of several factors for predicting PTBE after radiation treatment in meningioma patients. The optimal cut-off value was defined as the shortest distance from the upper left corner. The distance between each point on the ROC curve and the upper left corner was calculated as $\sqrt{(1 - \text{sensitivity})^2 + (1 - \text{specificity})^2}$ [15].

The cumulative hazard for PTBE was estimated using Kaplan-Meier analysis classified according to several predictive factors, with censoring of patients who had no PTBE on the last brain CT/MRI. Hazard ratios (HRs) with 95% confidence intervals (CIs) were then calculated using univariate and multivariate Cox regression analysis. This was used to identify independent predictive factors associated with the development of PTBE after LINAC-based radiation treatment in intracranial meningioma patients. The *P*-values less than 0.05 were considered statistically significant.

All statistical analyses were performed using R version 3.5.2 (https://www.r-project.org/).

# Results

## Characteristics of study patients

Seventy-Six patients with 83 intracranial meningiomas who received LINAC-based radiation treatments in our hospital over an approximate 5-year period were enrolled in the study. The mean patient age was 62.8 years and 80.7% of patients were female. The median imaging follow-up period was 456 days and 45.8% of patients had surgical resection before radiation treatment. The mean GTV and BED were 8.4 cc and 90.8 Gy, respectively. Non-PTBE and PTBE patients demonstrated significant differences in age. Details of patient characteristics are presented in Table 1.

## Mean frontal skull HU values, according to PTBE in study patients

Table 2 shows descriptive statistics of frontal skull HU values according to PTBE after radiation treatment.

We observed significant differences in values of the mean frontal skull HU and classification of the skull HU between non-PTBE and PTBE groups. The overall average mean frontal skull HU value was 725.8 in all study patients, 733.6 in the non-PTBE group and 547.8 in the PTBE group.

## Determination of the optimal cut-off values of predictive factors for PTBE after radiation

The optimal cut-off values of age, mean frontal skull HU, and GTV for the prediction of PTBE in patients with intracranial meningioma after radiation were 65 years (area under the curve [AUC] = 0.730; sensitivity = 84.6%; specificity = 65.7%; *P* = 0.009), 630.625 (AUC = 0.716; sensitivity = 76.9%; specificity = 67.1%; *P* = 0.014), and 7.194 cc (AUC = 0.706; sensitivity = 69.2%; specificity = 71.4%; *P* = 0.019), respectively (Fig 2A–2C).

However, BED did not show statistical significance in the ROC analysis (*P* = 0.920), (Fig 2D).

**Table 1. Characteristics of patients with intracranial meningioma who underwent LINAC-based radiation treatment in our hospital.**

| Characteristics | PTBE (-) | PTBE (+) | Total | P |
|---|---|---|---|---|
| Number (%) | 70 (84.3) | 13 (15.7) | 83 (100) | |
| Sex, female, n (%) | 56 (80.0) | 11 (84.6) | 67 (80.7) | 1.000 |
| Age, mean ± SD, y | 61.4 ± 11.6 | 70.2 ± 9.0 | 62.8 ± 11.7 | 0.012 |
| Imaging follow-up period, median (IQR), days | 477.0 (194.8–788.0) | 435.0 (198.5–1062.5) | 456.0 (198.0–862.0) | 0.251 |
| BMI, mean ± SD, kg/m$^2$ | 24.7 ± 3.7 | 24.2 ± 3.2 | 24.6 ± 3.6 | 0.675 |
| Height, mean ± SD, cm | 159.1 ± 9.4 | 155.9 ± 7.8 | 158.6 ± 9.2 | 0.247 |
| Weight, mean ± SD, kg | 62.5 ± 11.5 | 58.5 ± 6.7 | 61.9 ± 10.9 | 0.228 |
| Prior surgical resection, n (%) | 35 (50.0) | 3 (23.1) | 38 (45.8) | 0.128 |
| Pathology, n (%) | | | | 0.317 |
| WHO grade I | 24 (34.3) | 2 (15.4) | 26 (31.3) | |
| WHO grade II | 8 (11.4) | 0 | 8 (9.6) | |
| WHO grade III | 3 (4.3) | 1 (7.7) | 4 (4.8) | |
| GTV, mean ± SD, cc | 7.6 ± 9.9 | 12.4 ± 9.8 | 8.4 ± 10.0 | 0.116 |
| PTV, mean ± SD, cc | 11.7 ± 13.7 | 17.6 ± 11.1 | 12.7 ± 13.4 | 0.153 |
| Location, n (%) | | | | 0.733 |
| Convexity | 22 (31.4) | 5 (38.5) | 27 (32.5) | |
| Parasagittal or parafalcine | 14 (20.0) | 4 (30.8) | 18 (21.7) | |
| Sphenoid ridge | 7 (10.0) | 1 (7.7) | 8 (9.6) | |
| Cerebellopontine angle | 7 (10.0) | 2 (15.4) | 9 (10.8) | |
| Posterior fossa | 7 (10.0) | 1 (7.7) | 8 (9.6) | |
| Parasellar or petroclival | 10 (14.3) | 0 | 10 (12.0) | |
| Other | 3 (4.3) | 0 | 3 (3.6) | |
| Marginal radiation dose, mean ± SD, Gy | 31.5 ± 12.0 | 26.7 ± 5.6 | 30.8 ± 11.4 | 0.161 |
| Fractionation, n (%) | | | | 0.372 |
| SRS | 13 (18.6) | 3 (23.1) | 16 (19.3) | |
| hf-SRS (2–5 fractions) | 39 (55.7) | 9 (69.2) | 48 (57.8) | |
| hFSRT (6–10 fractions) | 4 (5.7) | 1 (7.7) | 5 (6.0) | |
| FSRT | 14 (20.0) | 0 | 14 (16.9) | |
| Dose per fraction median (IQR), Gy | 5.8 (4.8–7.0) | 6.0 (5.4–11.3) | 5.8 (5.3–7.0) | 0.418 |
| BED (α/β = 3), mean ± SD, Gy | 90.5 ± 16.2 | 92.5 ± 20.5 | 90.8 ± 16.8 | 0.695 |
| BED (α/β = 3), median (IQR), Gy | 86.4 (80.6–95.1) | 90.0 (75.6–102.1) | 86.4 (80.6–95.1) | 0.695 |
| Past medical history, n (%) | | | | |
| Hypertension | 29 (41.4) | 6 (46.2) | 35 (42.2) | 0.768 |
| Diabetes | 13 (18.6) | 3 (23.1) | 16 (19.3) | 0.708 |

LINAC, linear accelerator; PTBE, peritumoral brain edema; SD, standard deviation; IQR, interquartile range; BMI, body mass index; WHO, world health organization; GTV, gross tumor volume; PTV, planning target volume; SRS, stereotactic radiosurgery; hf-SRS, hypofractionated stereotactic radiosurgery; hFSRT, hypofractionated stereotactic radiotherapy; FSRT, fractionated stereotactic radiotherapy; BED, biologically equivalent dose

According to the cut-off values, the study patients were classified into (1) ≥65 years (2) mean frontal skull HU ≤630.625, and (3) GTV >7.194 cc groups.

## Cumulative hazard of PTBE after radiation according to several predictive factors

The incidence of PTBE was significantly higher among patients who were ≥65 years, with a mean frontal skull HU ≤630.625, and a GTV >7.194 cc in the clinical course of intracranial meningioma after LINAC-based radiation treatment (Fig 3A–3C).

**Table 2. Descriptive statistics of the mean frontal skull HU values according to peritumoral brain edema after LINAC-based radiation treatment in patients with intracranial meningioma.**

| Characteristics | PTBE (-) | PTBE (+) | Total | P |
|---|---|---|---|---|
| Overall mean frontal skull HU value, median (IQR) | 733.6 (559.3–870.1) | 547.8 (415.6–677.5) | 725.8 (527.0–853.3) | 0.018 |
| Overall mean frontal skull HU value, mean ± SD | 735.4 ± 246.2 | 564.4 ± 161.7 | 708.6 ± 242.4 | 0.018 |
| Mean HU value at each of four sites in the frontal skull, mean ± SD | | | | |
| Right lateral | 707.3 ± 245.1 | 579.2 ± 124.9 | 687.2 ± 234.6 | 0.070 |
| Right medial | 773.6 ± 268.7 | 588.9 ± 191.3 | 744.7 ± 265.9 | 0.021 |
| Left medial | 738.8 ± 271.4 | 566.2 ± 201.9 | 711.8 ± 268.2 | 0.032 |
| Left lateral | 722.1 ± 259.0 | 523.2 ± 166.3 | 690.9 ± 256.5 | 0.009 |
| Average, medial | 756.2 ± 266.0 | 577.5 ± 190.6 | 728.2 ± 262.9 | 0.024 |
| Average, lateral | 714.7 ± 247.9 | 551.2 ± 143.3 | 689.1 ± 241.4 | 0.024 |
| Classification of skull HU, n (%) | | | | 0.005 |
| Mean frontal skull HU ≤630.6 | 23 (32.9) | 10 (76.9) | 33 (39.8) | |
| Mean frontal skull HU >630.6 | 47 (67.1) | 3 (23.1) | 50 (60.2) | |

HU, Hounsfield unit; LINAC, linear accelerator; PTBE, peritumoral brain edema; IQR, interquartile range; SD, standard deviation

Patients with ≤5 fractionation (SRS or hf-SRS) also tended to have higher rates of PTBE after radiation (Fig 3D, *P* = 0.159).

## Independent predictive factors for PTBE after radiation in meningioma patients

The multivariate Cox regression analysis identified a mean frontal skull HU ≤630.625 and GTV >7.194 cc as independent predictors of PTBE after LINAC-based radiation treatment in intracranial meningioma patients (HR, 8.41; 95% CI, 1.42–49.83; *P* = 0.019; HR, 5.92; 95% CI, 1.16–30.19; *P* = 0.032, respectively); (Table 3).

Although we adjusted the age group in the multivariate analysis, a negative relationship between age and BMD may affect our results. We also identified a close association between age and mean frontal skull HU values in the study patients in S1 Fig. We further performed additional multivariate Cox regression with the adjustment for age as a continuous variable in the S1 Table. The results showed that the mean frontal skull HU ≤630.625 was maintained as an independent predictor of PTBE (HR, 7.04; 95% CI, 1.16–42.85; *P* = 0.034). When we adjusted for the past medical history, mean frontal skull HU ≤630.625 showed a strong association with PTBE in the study patients (S2 Table).

When the patients were divided into the risk factor group (age ≥65 years and skull HU ≤630.625 and GTV >7.194 cc) and others, the rate of PTBE was significantly higher in the risk factor group than in the others (Fig 4).

The univariate Cox analysis showed a strong significant association between PTBE and the risk factor group (HR, 21.92; 95% CI, 6.10 to 78.74; *P*<0.001).

## Discussion

We found that PTBE was independently associated with possible low BMD and large tumor volume in the clinical course of intracranial meningioma after LINAC-based radiation treatment. Older age showed a marginal independent association with PTBE occurrence after radiation. The possible low BMD group (mean skull HU ≤630.6) had an approximate 7.0 to 9.0-fold increased risk of PTBE after adjusting for other predictive factors including age. To

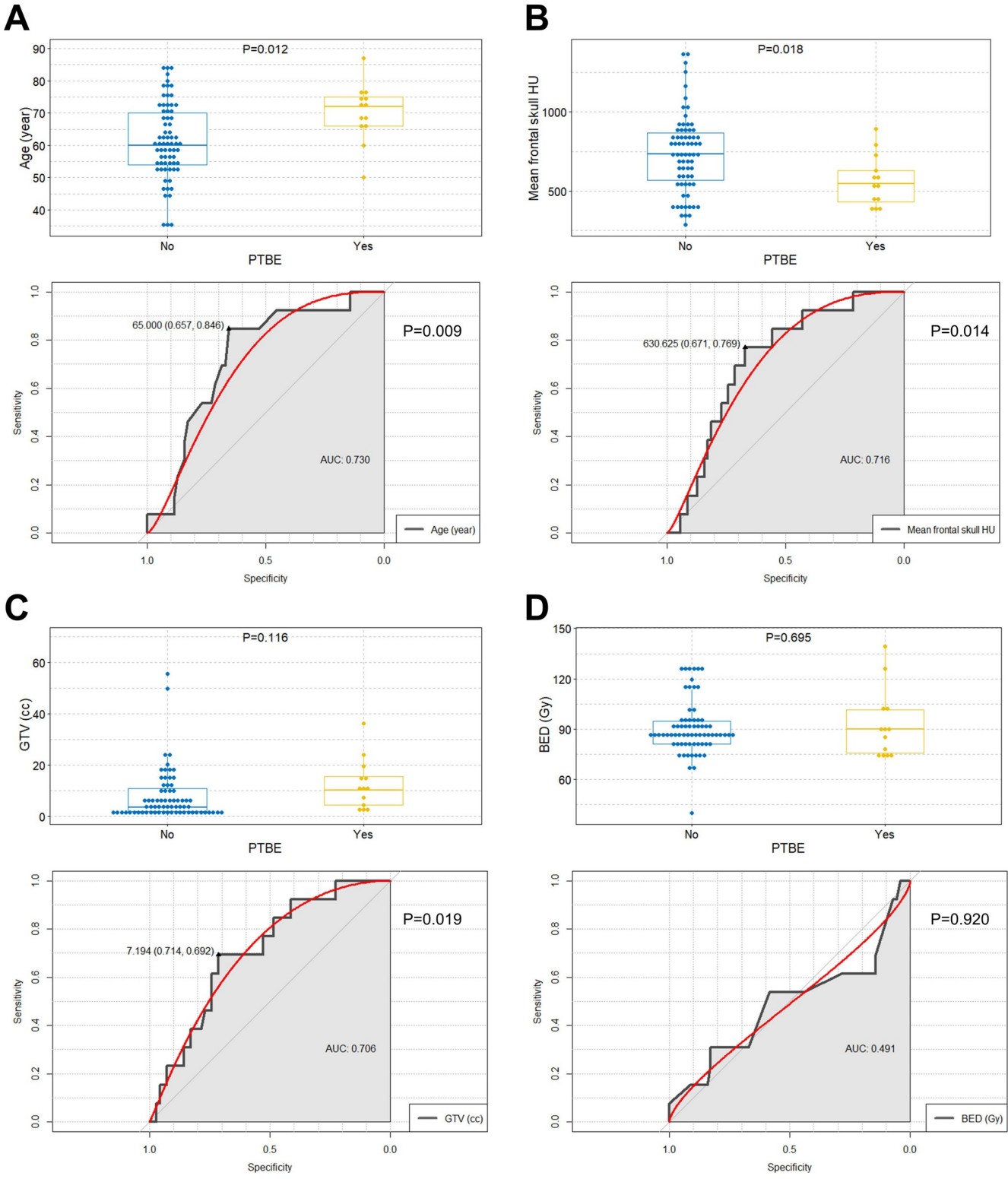

**Fig 2. Comparisons of age mean frontal skull HU value, GTV, and BED between PTBE and non-PTBE groups This includes the determination of the optimal cut-off values of the predictive factors for PTBE occurrence after radiation in intracranial meningioma.** (A) Boxplots with dot plots of age according to the PTBE and ROC curve to identify the optimal cutoff value of age for the prediction of PTBE; (B) Boxplots with dot plots of mean frontal skull HU according to the PTBE and ROC curve to identify the optimal cutoff value of mean frontal skull HU for the prediction of PTBE; (C) Boxplots with dot plots of GTV according to the PTBE and ROC curve to identify the optimal cutoff value of GTV for the prediction of PTBE; (D) Boxplots with dot plots

of BED according to the PTBE and ROC curve to identify the optimal cutoff value of BED for the prediction of PTBE. PTBE = peritumoral brain edema; AUC = area under the curve; HU = Hounsfield unit; GTV = gross tumor volume; BED = biologically equivalent dose; ROC = receiver operating characteristic.

our knowledge, this study is the first to suggest that BMD is associated with PTBE after radiation treatment in patients with intracranial meningioma.

It is well accepted that the tumor-brain barrier disruption may be an essential component of PTBE formation [6]. Glioblastoma and metastatic tumors usually induce PTBE. However,

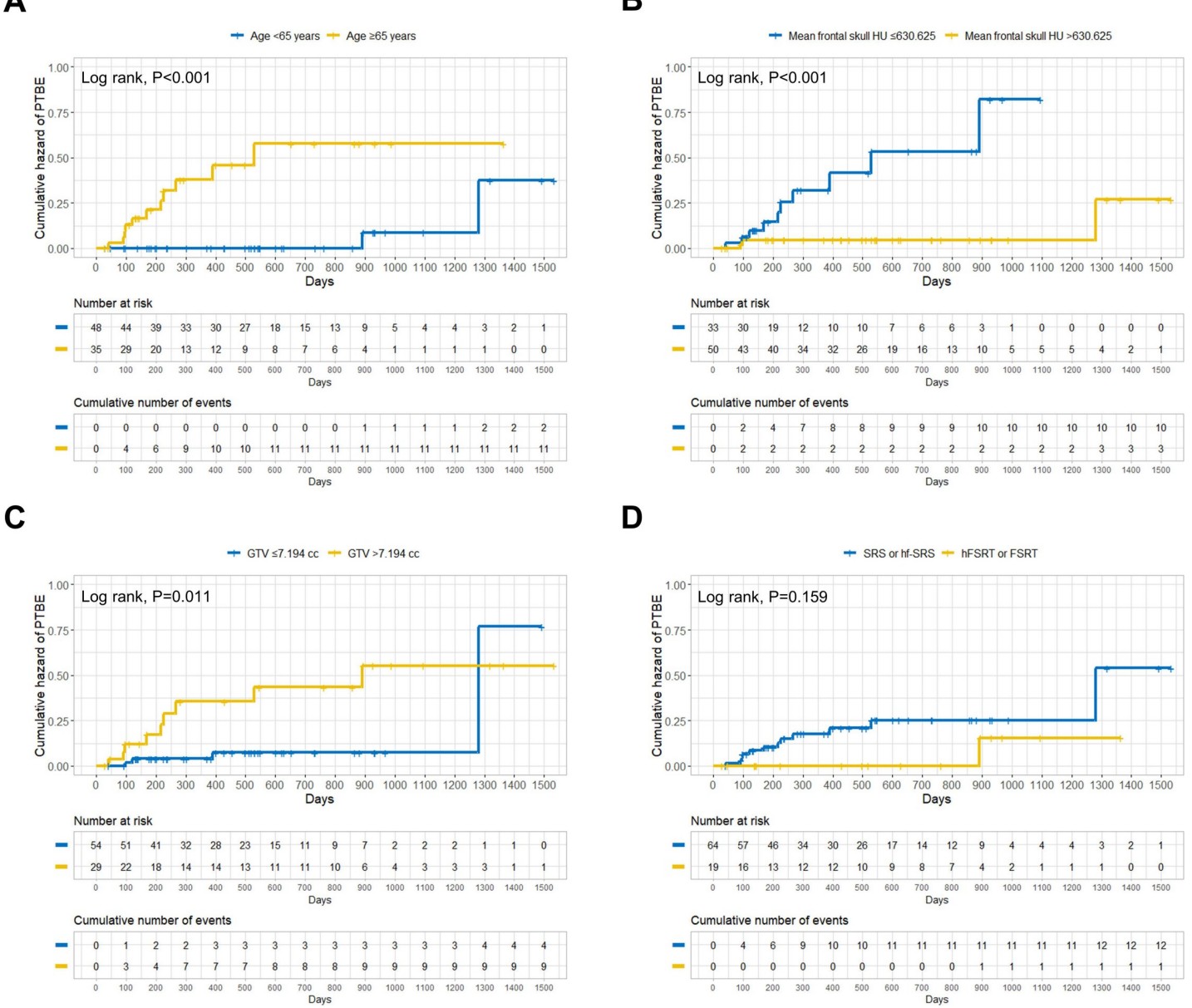

**Fig 3. Cumulative hazard of PTBE after LINAC-based radiation treatment for intracranial meningioma according to the several predictive factors.** (A) age group (cut-off value of 65); (B) mean frontal skull HU (cut-off value of 630.625); (C) GTV (cut-off value of 7.194); (D) two fractionation categories (SRS or hf-SRS versus hFSRT or FSRT). PTBE = peritumoral brain edema; HU = Hounsfield unit; GTV = gross tumor volume; SRS = stereotactic radiosurgery; hf-SRS = hypofractionated stereotactic radiosurgery; hFSRT = hypofractionated stereotactic radiotherapy; FSRT = fractionated stereotactic radiotherapy.

**Table 3. Univariate and multivariate Cox regression analyses for the development of peritumoral brain edema in patients with intracranial meningioma after LINAC-based radiation treatment based on predictive variables.**

| Variable | Univariate analysis | | Multivariate analysis | |
|---|---|---|---|---|
| | HR (95% CI) | *P* | HR (95% CI) | *P* |
| Sex | | | | |
| Male | Reference | | Reference | |
| Female | 1.27 (0.28–5.80) | 0.759 | 0.84 (0.17–4.26) | 0.836 |
| Age group | | | | |
| <65 years | Reference | | Reference | |
| ≥65 years | 11.24 (2.47–51.27) | 0.002 | 5.12 (0.99–26.65) | 0.052 |
| BMI (per 1 BMI increase) | 0.95 (0.80–1.14) | 0.610 | 0.97 (0.76–1.24) | 0.826 |
| Mean frontal skull HU | | | | |
| ≤630.6 | 9.83 (2.13–45.23) | 0.003 | 8.41 (1.42–49.83) | 0.019 |
| >630.6 | Reference | | Reference | |
| GTV | | | | |
| ≤7.2 cc | Reference | | Reference | |
| >7.2 cc | 4.17 (1.27–13.74) | 0.019 | 5.92 (1.16–30.19) | 0.032 |
| Location | | | | |
| Convexity | 2.41 (0.74–7.88) | 0.145 | 1.90 (0.52–7.02) | 0.333 |
| Other regions | Reference | | Reference | |
| BED (α/β = 3) (per 1 Gy increase) | 1.01 (0.97–1.04) | 0.725 | 1.00 (0.95–1.05) | 0.989 |
| Fractionation (per 1 fraction increase) | 0.92 (0.82–1.04) | 0.184 | 0.78 (0.45–1.36) | 0.381 |

HR, hazard ratio; CI, confidence interval; BMI, body mass index; HU, Hounsfield unit; GTV, gross tumor volume; BED, biologically equivalent dose

Patients who were ≥65 years showed a marginal statistically significant association with PTBE occurrence after full adjustment (HR, 5.12; 95% CI, 0.99–26.65; *P* = 0.052).

in contrast to glioblastomas and metastases, meningiomas are encapsulated and are separated from the underlying normal cerebral cortex by the arachnoid membrane and pia mater. The arachnoid membrane is impermeable to fluids due to its' tight intercellular junctions [16]. It is thought that the arachnoid membrane may act as a mechanical and biochemical buffer against mediators released from a meningioma [17]. It is probable that the arachnoid membrane blocks the spread of edema-associated proteins such as endothelial growth factor/vascular permeability factor and vasogenic edema fluids from meningiomas from the peritumoral brain tissue [3]. A previous study that examined the microscopic anatomy of the brain-meningioma interface, also reported that the degree of arachnoid disruption correlated with the presence of perifocal edema [7].

Interestingly, a microscopic examination of the brain-meningioma interface revealed proliferation of hyperplastic arachnoid trabeculae, (below the arachnoid membrane at the brain–meningioma interface) in the meningioma with a thin connective capsule (shown in Fig 1A of the study) [7]. After the study, it was reported that the arachnoid trabeculae and granulations are composed of type 1 collagen [18]. The arachnoid is composed of two layers. An outer part of the arachnoid is the arachnoid barrier layer and is an actual membrane cover. An inner part is the arachnoid trabeculae maintaining the stability of the subarachnoid space and cerebrospinal fluid flow to support the arachnoid barrier layer [19]. Arachnoid cap cells are believed to be of meningioma cell origin [20]. Therefore, it is possible to postulate that meningioma from arachnoid cap cells may naturally push the arachnoid trabeculae into the pia mater [21]. As the tumor grows, it could also be assumed that arachnoid trabeculae may be sandwiched between the pia mater and meningioma. This may form part of the tumor-brain contact interface.

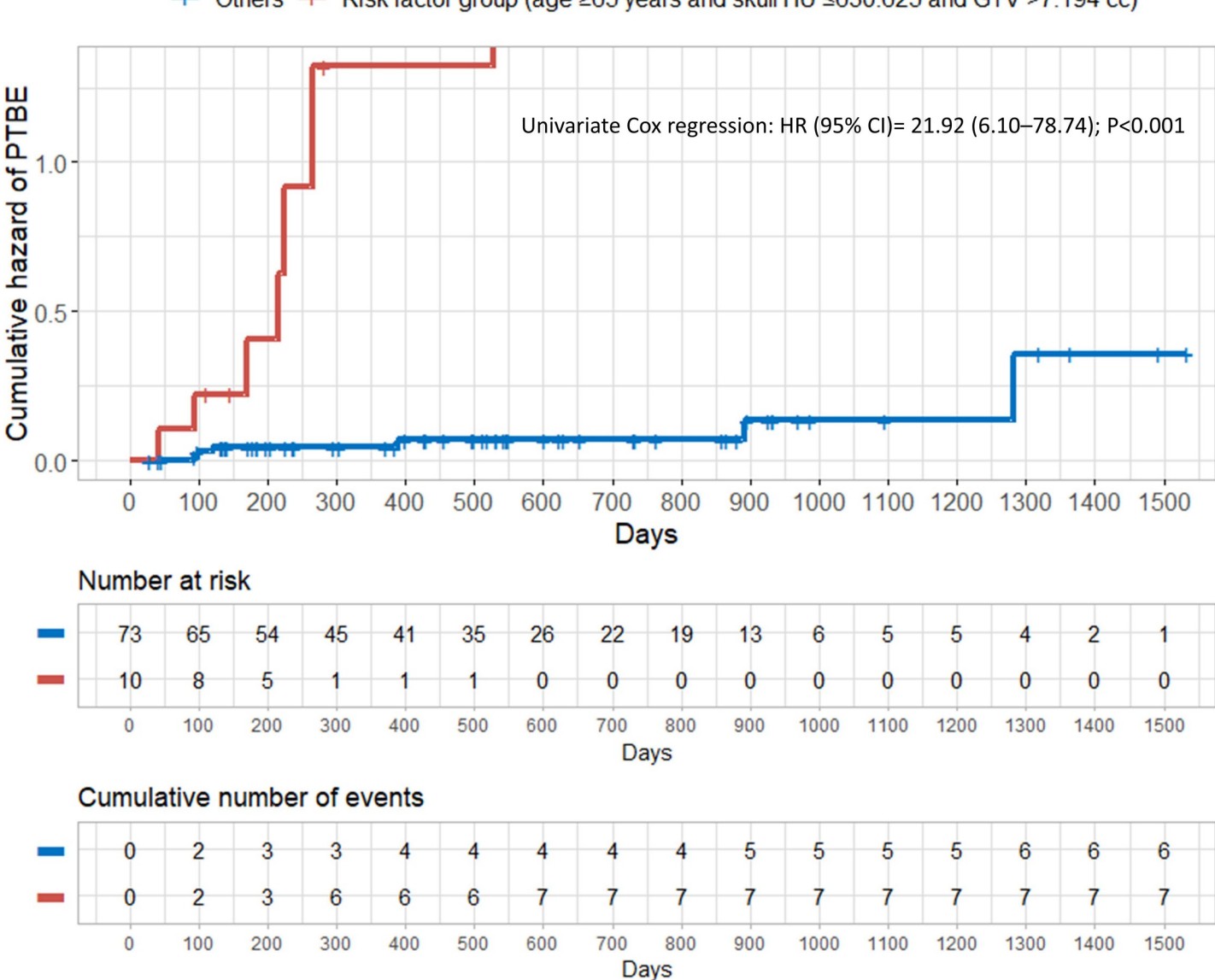

**Fig 4. Cumulative hazard of PTBE after LINAC-based radiation treatment for intracranial meningioma according to the risk factor group (age ≥65 years, skull HU ≤630.625 and GTV >7.194 cc).** PTBE = peritumoral brain edema; HU = Hounsfield unit; GTV = gross tumor volume; HR = hazard ratio; CI = confidence interval.

Compression due to the growth of a tumor on adjacent venous structures, leptomeninges, and the cerebral cortex may lead to an increase in hydrostatic pressure [22].

It is well documented that type 1 collagen is a major component of bone. Osteoporosis is a systemic disease that affects systemic bone mass and microarchitecture throughout the body. We previously reported the close association between mean frontal skull HU and BMD [8,14]. We also demonstrated that systemic osteoporosis may negatively affect the integrity of arachnoid trabeculae and granulations because bone, arachnoid trabeculae, and granulations are all composed of type 1 collagen [8,9]. Supporting our hypothesis, osteogenesis imperfecta, that is caused by mutations in type 1 procollagen genes (*COL1A1/COL1A2*), is associated with communicating hydrocephalus [23]. We believe that trabeculae, which are sandwiched between

the pia mater and meningioma, may be more impaired and weakened in osteoporotic patients when compared to healthy patients.

Previous studies described that irradiation affects collagen structure and can lead to collagen changes and damage [24,25]. When the meningioma is not treated with surgery or radiation therapy, tumor growth is the primary cause of damage to the tumor-brain contact interface including the arachnoid trabeculae. After radiation, this contact interface may be damaged by radiation activities [3].

Based on the above findings and assumptions, we propose the following hypothetical mechanism as an explanation for the association between possible low BMD, large tumor volume, and PTBE after radiation for intracranial meningioma. As tumor grows, the tumor may push more of the arachnoid trabeculae into the pia mater and cause damage to the tumor-brain contact interface. The larger the tumor, the greater the likelihood of damage to the tumor-brain contact interface including the arachnoid trabeculae. The damage to the arachnoid trabeculae due to compression by the tumor will be more severe in osteoporotic patients. Radiation may aggravate the damaged tumor-brain contact interface including the arachnoid trabeculae and may lead to tumor-brain barrier disruption. We hypothesized that the more damaged the arachnoid trabeculae are at the tumor-brain interface due to low BMD and large tumor volume, the higher the possibility will be of tumor-brain barrier disruption after radiation therapy. Tumor-brain barrier disruption may result in PTBE formation in meningioma patients after radiation.

Loosening of the microstructure network and the volume reduction of aging white matter may increase the possibility of PTBE. This allows direct transmission of edematous fluids into the white matter [26]. We believe that thorough precautions are required with older patients with osteoporosis and large tumor volume, after radiation therapy for intracranial meningioma. We also found that BED was not associated with PTBE occurrence in our study. We propose that this was because we did not use extremely high radiation doses and the narrow BED range may not have resulted in significant differences in PTBE occurrence [3]. We believe that the status of the brain-meningioma contact interface, including the arachnoid trabeculae, is a more important factor than the radiation dosage in predicting PTBE occurrence after radiation for meningioma. Although it falls short of significance, multi-fraction seems to be important for prevention of PTBE after radiation for meningioma.

Our study has several limitations. First, due to the retrospective nature of the study, the length of follow-ups and the number of follow-up images varied widely. Second, HU measurement errors may have occurred. However, all brain CT images were magnified for HU measurement to reduce errors. We excluded patients with no measurable cancellous bone of the frontal skull in the simulation brain CT. To reduce measurement errors, we estimated mean HU values in four areas of the frontal skull and averaged them. Third, although HU values are correlated with BMD, HU values may not reflect the exact BMD values. Fourth, heterogeneity in tumor location and absence of histological confirmation in many cases may affect the results of the study. Fifth, several previously reported risk factors for PTBE after radiosurgery for meningioma were not included for the study. These include adjacency to vein or sinus, tight vs smooth brain-tumor interface, plasma levels of vascular endothelial growth factor, or a few others [5]. Lastly, the small number of cases may have reduced the statistical power and validation. In addition, our single-center study may be limited in its broad applicability. A further multi-center study with a larger sample size would be required to validate our results.

In conclusion, our study suggests that possible osteoporotic conditions, large tumor volume, and older age may be associated with PTBE occurrence after LINAC-based radiation treatment for intracranial meningioma. We believe that these findings may be helpful for predicting PTBE occurrence during the clinical course of meningioma after radiation. In the

future, we anticipate that the findings of this study may enhance the understanding of the underlying mechanisms of PTBE after radiation in meningioma patients.

## Supporting information

**S1 Fig. Scatterplot with linear regression line showing the association between age and mean frontal skull HU values.** HU = Hounsfield unit.
(TIF)

**S1 Table. Uni- and multivariate Cox regression analyses for the development of peritumoral brain edema in patients with intracranial meningioma after LINAC-based radiation treatment based on predictive variables (adjusted for age as continuous variable).**
(DOCX)

**S2 Table. Uni- and multivariate Cox regression analyses for the development of peritumoral brain edema in patients with intracranial meningioma after LINAC-based radiation treatment based on predictive variables (adjusted for age as continuous variable and past medical history).**
(DOCX)

## Author Contributions

**Conceptualization:** Myung-Hoon Han.

**Data curation:** Ryang-Hun Lee.

**Formal analysis:** Myung-Hoon Han.

**Investigation:** Ryang-Hun Lee.

**Methodology:** Myung-Hoon Han.

**Resources:** Jin Hwan Cheong.

**Supervision:** Jae Min Kim, Jin Hwan Cheong, Je Il Ryu, Young Soo Kim.

**Visualization:** Myung-Hoon Han.

**Writing – original draft:** Ryang-Hun Lee, Myung-Hoon Han.

**Writing – review & editing:** Je Il Ryu.

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
