## [Decision Letter · Decision Letter 0]

22 Jan 2020

PONE-D-19-32359

Predictive factors for the development of peritumoral brain edema after LINAC-based radiation treatment in patients with intracranial meningioma

PLOS ONE

Dear Prof Han,

Thank you for submitting your manuscript to PLOS ONE. After careful consideration, we feel that it has merit but does not fully meet PLOS ONE’s publication criteria as it currently stands. Therefore, we invite you to submit a revised version of the manuscript that addresses the points raised during the review process.

We would appreciate receiving your revised manuscript by Mar 07 2020 11:59PM. To enhance the reproducibility of your results, we recommend that if applicable you deposit your laboratory protocols in protocols.io, where a protocol can be assigned its own identifier (DOI) such that it can be cited independently in the future. For instructions see: http://journals.plos.org/plosone/s/submission-guidelines#loc-laboratory-protocols

We look forward to receiving your revised manuscript.

Kind regards,

Jonathan H Sherman

Academic Editor

PLOS ONE

Journal Requirements:

Reviewers' comments:

Reviewer's Responses to Questions

**Comments to the Author**

1. Is the manuscript technically sound, and do the data support the conclusions?

Reviewer #1: Yes

Reviewer #2: Yes

2. Has the statistical analysis been performed appropriately and rigorously? 

Reviewer #1: Yes

Reviewer #2: Yes

3. Have the authors made all data underlying the findings in their manuscript fully available?

Reviewer #1: Yes

Reviewer #2: Yes

4. Is the manuscript presented in an intelligible fashion and written in standard English?

Reviewer #1: No

Reviewer #2: Yes

5. Review Comments to the Author

Reviewer #1: 1. Please comment on the authors' question and originality of their findings.

Intriguing question, with relatively original findings.

2. Please comment on the appropriateness of the study approach and experimental design. (Examples: retrospective or prospective cohort, case-control, cross-sectional, ecological, case series; clinical trial or secondary analysis of clinical trial; registry-based; critical review; metaanalysis or systematic review; experimental, based on cell cultures, animal models, physical models, or method/technique development.)

Appears to be a single-center study, limiting its broad applicability.

3. Please comment on the appropriateness and reproducibility of the data collection and experimental techniques. (If applicable, does the study comply with the CONSORT, PRISMA and/or REMARK statements? If applicable, was the study IRB-approved or registered on clinicaltrials.gov?)

The authors claim this study was IRB-approved. The methods and analysis described is appropriate and reproducible. I suspect other groups will rapidly run similar analyses on databases in an attempt to validate the findings reported here.

4. Please comment on the analysis and interpretations of the data. Do you agree with the proposed conclusions?

Not yet -- please see my suggestions, particularly the first comment.

5. Please comment on weaknesses or limitations of the study. (Examples are: selection biases, sample size limitations, missing data.)

Given the relatively small sample size, these preliminary results are intriguing but require validation in a larger cohort. The authors acknowledge this.

6. Please comment on the writing and organization of the paper. Is the paper overly wordy? Is the English language acceptable?

Well-written.

7. Please comment on the necessity and clarity of the figures and tables. Can they stand independently of the text?

They support the manuscript.

8. Please comment on any need for formal statistical review.

Always useful.

Reviewer #2: The authors present data from 76 patients and 83 meningioma lesions treated with LINAC based radiation over a 5 year period. They present several factors correlated to the development of peri-tumoral brain edema (PTBE) after radiation. Their study finds a significant and independent association of low frontal skull hounsfeld units (HU) to the development of PTBE in addition to well known factors such as tumor volume. The association of frontal skull HU is a novel finding and to my knowledge, not previously reported or studied in the literature.

I request the following edits and clarifications:

Title:

1) The title is generic and indistinguishable from many other articles on this topic, and if frontal skull HU is a significant finding, then I suggest making the title more specific. One possible title is "Significance of skull osteoporosis to the development of peritumoral brain edema after LINAC-based radiation treatment in patients with intracranial meningioma" - however, I leave this at the author's discretion.

Abstract:

Ok. No additional comments.

Introduction:

A major review of the topic of radiation induced edmea was published by Milano et al in 2018 in IJROBP (Pubmed ID 29726362). It may be helpful to cite this review and discuss the features identified in this review article that were significant for PTBE and how frontal bone HU was not previously reported as a relevant factor in this and other reviews.

Methods:

Study Patients: ok

Radiation Technique:

I am not sure that the alpha/beta ratio of 10 for calculating BED for meningiomas and for estimating toxicity is accurate in this paper. For instance, it is known that meningioma, being a benign tumor, has an alpha/beta ratio of closer to 3.7 (Pubmed ID 20470198) Furthermore, this study relates to the effects of radiation on normal brain tissue since the endpoint is edema. Normal CNS tissue is thought to have an alpha/beta ratio of close to 2 (pubmed ID 3891621 among others) although using a value of 3 is also acceptable. I believe the BED estimates in this paper need to be re-calculated with these more appropriate values (select 2 or 3 for normal brain, and select 3 or 3.7 for meningioma). As an example of a paper that used alpha/beta=3, see Pubmed ID 18812954.

Measurement of frontal skull HU:

1) As the mean HU value is acquired from PACS, the PACS software name and version should be reported in the methods of the paper.

Statistics: OK

Results:

Characteristics of study patients: OK.

Mean frontal skull HU values: OK.

Determination of optimal cut off values: OK.

Cumulative hazard: OK, but see figure 3 comment below.

Independent predictive factors: Ok.

Discussion:

Previously reported risk factors for PTBE have included tumor location, marginal dose, adjacency to vein or sinus, tight vs smooth brain-tumor interface, and a few others per review in Pubmed ID 29726362. Although this was not the focus of the paper, can you hypothesize why your paper identified your factors and not others as contributing to PTBE?

Tables:

Table 1: OK

Table 2: OK

Figures:

Figure 1: Ok.

Figure 2: Ok.

Figure 3: All of the cumulative hazard graphs have different ranges (roughly between 0.5 and 0.8). Please have the cumulative hazard graphs all have a scale of 0 to 1.0 for more reasonable visual comparison.

Figure 4: The cumulative hazard appears to go above 1.0 in the risk factor group. Please explain as there may be an error in the analysis that results in >100% of lesions in this group having an event.

Supplement figures and tables: Ok.

6. PLOS authors have the option to publish the peer review history of their article (what does this mean?). If published, this will include your full peer review and any attached files.

Reviewer #1: No

Reviewer #2: Yes: Yuan James Rao

---

## [Author Response · Author response to Decision Letter 0]

28 Jan 2020

Response to reviewers 

Significance of skull osteoporosis to the development of peritumoral brain edema after LINAC-based radiation treatment in patients with intracranial meningioma

Ryang-Hun Lee, Jae Min Kim, Jin Hwan Cheong, Je Il Ryu, Young Soo Kim, Myung-Hoon Han

Reviewer #1 

2. Please comment on the appropriateness of the study approach and experimental design. (Examples: retrospective or prospective cohort, case-control, cross-sectional, ecological, case series; clinical trial or secondary analysis of clinical trial; registry-based; critical review; metaanalysis or systematic review; experimental, based on cell cultures, animal models, physical models, or method/technique development.)

Appears to be a single-center study, limiting its broad applicability. 

4. Please comment on the analysis and interpretations of the data. Do you agree with the proposed conclusions?

Not yet -- please see my suggestions, particularly the first comment. 

5. Please comment on weaknesses or limitations of the study. (Examples are: selection biases, sample size limitations, missing data.)

Given the relatively small sample size, these preliminary results are intriguing but require validation in a larger cohort. The authors acknowledge this.

Thank you. We agree with your comment. We also agree that the hypothesis of our study requires validation in a larger cohort as indicated by you. We have added relevant sentences in the limitations section of the Discussion as follows:

Discussion

Lastly, the small number of cases may have reduced the statistical power and validation. In addition, our single-center study may be limited in its broad applicability. A further multi-center study with a larger sample size would be required to validate our results.

Reviewer #2

1. Title:

1) The title is generic and indistinguishable from many other articles on this topic, and if frontal skull HU is a significant finding, then I suggest making the title more specific. One possible title is "Significance of skull osteoporosis to the development of peritumoral brain edema after LINAC-based radiation treatment in patients with intracranial meningioma" - however, I leave this at the author's discretion. 

Thank you. We agree with your comment. We have revised the title based on your suggestion as follows:

Title: Significance of skull osteoporosis to the development of peritumoral brain edema after LINAC-based radiation treatment in patients with intracranial meningioma Predictive factors for the development of peritumoral brain edema after LINAC-based radiation treatment in patients with intracranial meningioma

2. Introduction:

A major review of the topic of radiation induced edmea was published by Milano et al in 2018 in IJROBP (Pubmed ID 29726362). It may be helpful to cite this review and discuss the features identified in this review article that were significant for PTBE and how frontal bone HU was not previously reported as a relevant factor in this and other reviews.

Thank you for your valuable comment. We have cited and discussed the review (Pubmed ID 29726362) in the Introduction as suggested by you. Please find the revised text below:

Introduction

It was reported that symptomatic brain edema occurs in 37.5% of patients with parasagittal meningiomas after gamma knife radiosurgery [4]. Previously, several risk factors associated with peritumoral brain edema (PTBE) after radiosurgery in meningioma were reported. These include greater radiation dose, greater tumor size or volume, tumor location, brain-tumor interface, no prior resection for meningioma, and presence of pretreatment edema [5]. 

We hypothesized that osteoporotic conditions may be associated with PTBE after radiation in intracranial meningioma patients. To the best of our knowledge, there are no previous studies describing the possible relationship between osteoporotic conditions and PTBE after radiotherapy in meningioma which have been published [5].

3. Methods:

Study Patients: ok

Radiation Technique:

I am not sure that the alpha/beta ratio of 10 for calculating BED for meningiomas and for estimating toxicity is accurate in this paper. For instance, it is known that meningioma, being a benign tumor, has an alpha/beta ratio of closer to 3.7 (Pubmed ID 20470198) Furthermore, this study relates to the effects of radiation on normal brain tissue since the endpoint is edema. Normal CNS tissue is thought to have an alpha/beta ratio of close to 2 (pubmed ID 3891621 among others) although using a value of 3 is also acceptable. I believe the BED estimates in this paper need to be re-calculated with these more appropriate values (select 2 or 3 for normal brain, and select 3 or 3.7 for meningioma). As an example of a paper that used alpha/beta=3, see Pubmed ID 18812954.

We agree with your comment. We re-calculated the BED with α/β=3 and have revised the relevant Tables, Figures, and content of the main text as shown below. We also cited the paper (Pubmed ID 18812954) in the Methods section.

Table 1.

BED (α/β =310), mean ± SD, Gy 90.5 ± 16.2 49.2 ± 8.7 92.5 ± 20.5 46.4 ± 4.7 90.8 ± 16.8 48.8 ± 8.2 0.695

0.264

BED (α/β =310), median (IQR), Gy 86.4 

(80.6–95.1)

46.8

(44.5–52.7) 90.0 

(75.6–102.1)

45.8

(41.6–49.2) 86.4 

(80.6–95.1)

45.9

(43.7–51.2) 0.695 0.264

Table 3.

 Univariate analysis Multivariate analysis

Variable HR (95% CI) P HR (95% CI) P 

Sex 

Male Reference Reference 

Female 1.27 (0.28–5.80) 0.759 0.84 (0.17–4.26) 0.83 (0.16–4.16) 0.836 0.818

Age group 

<65 years Reference Reference 

≥65 years 11.24 (2.47–51.27) 0.002 5.12 (0.99–26.65) 5.20 (1.00–27.13) 0.052 0.050

BMI (per 1 BMI increase) 0.95 (0.80–1.14) 0.610 0.97 (0.76–1.24) 0.98 (0.77–1.26) 0.826 0.893

Mean frontal skull HU 

≤630.6 9.83 (2.13–45.23) 0.003 8.41 (1.42–49.83) 8.38 (1.38–50.73) 0.019 0.021

>630.6 Reference Reference 

GTV 

≤7.2 cc Reference Reference 

>7.2 cc 4.17 (1.27–13.74) 0.019 5.92 (1.16–30.19) 5.78 (1.14–29.39) 0.032 0.034

Location 

Convexity 2.41 (0.74–7.88) 0.145 1.90 (0.52–7.02) 1.96 (0.53–7.23) 0.333 0.310

Other regions Reference Reference 

BED (α/β=310) 

(per 1 Gy increase) 1.01 (0.97–1.04) 0.96 (0.89–1.04) 0.725 0.352 1.00 (0.95–1.05) 0.97 (0.85–1.11) 0.989 0.688

Fractionation 

(per 1 fraction increase) 0.92 (0.82–1.04) 0.184 0.78 (0.45–1.36) 0.77 (0.49–1.20) 0.381 0.240

S1 Table.

 Univariate analysis Multivariate analysis

Variable HR (95% CI) P value HR (95% CI) P value

Sex 

Male Reference Reference 

Female 1.27 (0.28–5.80) 0.759 0.96 (0.18–5.30) 0.90 (0.16–5.03) 0.967 0.908

Age (per 1-year increase) 1.09 (1.03–1.16) 0.002 1.05 (0.97–1.13) 1.05 (0.97–1.14) 0.229 0.212

BMI (per 1 BMI increase) 0.95 (0.80–1.14) 0.610 0.98 (0.77–1.25) 0.99 (0.78–1.27) 0.889 0.952

Mean frontal skull HU 

≤630.6 9.83 (2.13–45.23) 0.003 7.04 (1.16–42.85) 6.99 (1.12–43.60) 0.034 0.037

>630.6 Reference Reference 

GTV 

≤7.2 cc Reference Reference 

>7.2 cc 4.17 (1.27–13.74) 0.019 7.20 (1.37–37.81) 7.19 (1.39–37.35) 0.020 0.019

Location 

Convexity 2.41 (0.74–7.88) 0.145 1.89 (0.50–7.11) 1.92 (0.51–7.18) 0.348 0.333

 Other regions Reference Reference 

BED (α/β=310) 

(per 1-Gy increase) 1.01 (0.97–1.04) 0.96 (0.89–1.04) 0.725 0.352 1.00 (0.95–1.05) 0.97 (0.85–1.10) 0.888 0.613

Fractionation 

(per 1-fraction increase) 0.92 (0.82–1.04) 0.184 0.72 (0.43–1.22) 0.73 (0.48–1.11) 0.221 0.140

S2 Table.

 Univariate analysis Multivariate analysis

Variable HR (95% CI) P value HR (95% CI) P value

Sex 

Male Reference Reference 

Female 1.27 (0.28–5.80) 0.759 1.06 (0.19–6.02) 1.03 (0.18–5.94) 0.950 0.974

Age (per 1-year increase) 1.09 (1.03–1.16) 0.002 1.06 (0.98–1.15) 1.06 (0.98–1.15) 0.130 0.139

BMI (per 1 BMI increase) 0.95 (0.80–1.14) 0.610 1.03 (0.80–1.33) 1.03 (0.80–1.33) 0.814 0.820

Mean frontal skull HU 

≤630.6 9.83 (2.13–45.23) 0.003 9.41 (1.45–61.16) 9.43 (1.41–62.86) 0.019 0.020

>630.6 Reference Reference 

GTV 

≤7.2 cc Reference Reference 

>7.2 cc 4.17 (1.27–13.74) 0.019 6.02 (1.06–34.06) 5.80 (1.04–32.23) 0.042 0.045

Location 

Convexity 2.41 (0.74–7.88) 0.145 1.65 (0.43–6.35) 1.75 (0.46–6.60) 0.465 0.412

 Other regions Reference Reference 

BED (α/β=310) 

(per 1-Gy increase) 1.01 (0.97–1.04) 0.96 (0.89–1.04) 0.725 0.352 1.01 (0.96–1.06) 1.00 (0.88–1.15) 0.646 0.948

Fractionation 

(per 1-fraction increase) 0.92 (0.82–1.04) 0.184 0.77 (0.45–1.31) 0.71 (0.45–1.12) 0.326 0.137

Past medical history 

Hypertension 1.12 (0.37–3.39) 0.837 0.44 (0.10–1.88) 0.48 (0.11–2.10) 0.266 0.326

Diabetes 1.12 (0.30–4.12) 0.870 0.57 (0.12–2.73) 0.61 (0.13–2.86) 0.483 0.531

Figure 2.

Abstract

Results: 

We found mean frontal skull HU ≤630.625 and gross tumor volume >7.194 cc to be independent predictors of PTBE after radiation treatment in patients with meningioma (hazard ratio, 8.418.38; P=0.0190.021; hazard ratio, 5.925.78; P=0.0320.034, respectively). In addition, patients who were ≥65 years showed a marginally significant association with PTBE.

Methods 

Radiation technique

The biologically equivalent dose (BED) for the tumor was calculated according to the following equation: BED = nd × (1 + d/10), where n is the number of fractions, d is the dose per fraction, and α/β=310 [12].

Results

Characteristics of study patients 

The mean GTV and BED were 8.4 cc and 90.848.8 Gy, respectively.

Determination of the optimal cut-off values of predictive factors for PTBE after radiation

However, BED did not show statistical significance in the ROC analysis (P=0.9200.335), (Fig. 2D). 

Independent predictive factors for PTBE after radiation in meningioma patients

The multivariate Cox regression analysis identified a mean frontal skull HU ≤630.625 and GTV >7.194 cc as independent predictors of PTBE after LINAC-based radiation treatment in intracranial meningioma patients (HR, 8.418.38; 95% CI, 1.42–49.831.38–50.73; P=0.0190.021; HR, 5.925.78; 95% CI, 1.16–30.191.14–29.39; P=0.0320.034, respectively); (Table 3). Patients who were ≥65 years showed a marginal statistically significant association with PTBE occurrence after full adjustment (HR, 5.125.20; 95% CI, 0.99–26.651.00–27.13; P=0.0520.050).

We further performed additional multivariate Cox regression with the adjustment for age as a continuous variable in the S1 Table. The results showed that the mean frontal skull HU ≤630.625 was maintained as an independent predictor of PTBE (HR, 7.046.99; 95% CI, 1.16–42.851.12–43.60; P=0.0340.037). When we adjusted for the past medical history, mean frontal skull HU ≤630.625 showed a strong association with PTBE in the study patients (S2 Table).

4. Measurement of frontal skull HU:

1) As the mean HU value is acquired from PACS, the PACS software name and version should be reported in the methods of the paper.

We added a relevant sentence in the Methods section as per your suggestion. The revised text is as follows:

Methods 

Measurement of frontal skull HU

The HU value of the frontal cancellous bone was measured using the “Linear histogram graph” function in the picture archiving and communication system (PACS) (PiViewSTAR version 5.0, INFINITT Healthcare, Seoul, Korea).

5. Discussion:

Previously reported risk factors for PTBE have included tumor location, marginal dose, adjacency to vein or sinus, tight vs smooth brain-tumor interface, and a few others per review in Pubmed ID 29726362. Although this was not the focus of the paper, can you hypothesize why your paper identified your factors and not others as contributing to PTBE?

We appreciate your comment. We previously described the limitation regarding heterogeneity in tumor location due to small sample size in the Discussion section as follows:

Discussion

Fourth, heterogeneity in tumor location and absence of histological confirmation in many cases may affect the results of the study.

Although short of achieving statistical significance, Table 3 shows a tendency towards higher PTBE occurrence in the convexity meningioma as compared to meningiomas in other locations.

Table 3.

Location 

Convexity 2.41 (0.74–7.88) 0.145 1.90 (0.52–7.02) 1.96 (0.53–7.23) 0.333 0.310

Other regions Reference Reference 

However, we agree that the hypothesis of our study requires validation in a larger cohort with consideration to tumor location. In addition, marginal radiation dose was not a risk factor for PTBE after radiation in meningioma in the study. We also previously discussed a possible reason in the Discussion section as follows:

Discussion

We also found that BED was not associated with PTBE occurrence in our study. We propose that this was because we did not use extremely high radiation doses and the narrow BED range may not have resulted in significant differences in PTBE occurrence [3]. We believe that the status of the brain-meningioma contact interface, including the arachnoid trabeculae, is a more important factor than the radiation dosage in predicting PTBE occurrence after radiation for meningioma.

Other reported risk factors were not included for the study. Therefore, we added a relevant sentence and cited a review article (Pubmed ID 29726362) in the limitations section of the Discussion section as follows:

Discussion

Fourth, heterogeneity in tumor location and absence of histological confirmation in many cases may affect the results of the study. Fifth, several previously reported risk factors for PTBE after radiosurgery for meningioma were not included for the study. These include adjacency to vein or sinus, tight vs smooth brain-tumor interface, plasma levels of vascular endothelial growth factor, or a few others [5]. 

6. Figures:

Figure 3: All of the cumulative hazard graphs have different ranges (roughly between 0.5 and 0.8). Please have the cumulative hazard graphs all have a scale of 0 to 1.0 for more reasonable visual comparison.

We agree with your comment. We have adjusted the scale of the y-axes to be (0.0 to 1.0) of the Figure 3 as follows:

Figure 3.

7. Figure 4: The cumulative hazard appears to go above 1.0 in the risk factor group. Please explain as there may be an error in the analysis that results in >100% of lesions in this group having an event.

We appreciate your comment. However, it is known that the cumulative hazard can go above 1.0 according to previous studies [1–3].

1. Bybee KA, Lee JH, O’Keefe JH. Cumulative clinical trial data on atorvastatin for reducing cardiovascular events: the clinical impact of atorvastatin. Curr Med Res Opin. 2008;24: 1217–1229. doi:10.1185/030079908x292001

2. Colhoun HM, Betteridge DJ, Durrington PN, Hitman GA, Neil HAW, Livingstone SJ, et al. Primary prevention of cardiovascular disease with atorvastatin in type 2 diabetes in the Collaborative Atorvastatin Diabetes Study (CARDS): multicentre randomised placebo-controlled trial. Lancet Lond Engl. 2004;364: 685–696. doi:10.1016/S0140-6736(04)16895-5

3. Jahn-Eimermacher A, Ingel K, Ozga A-K, Preussler S, Binder H. Simulating recurrent event data with hazard functions defined on a total time scale. BMC Med Res Methodol. 2015;15: 16. doi:10.1186/s12874-015-0005-2

We estimated the cumulative hazard using the ggplot2 and ggsurvplot functions of R version 3.5.2 as described in the Methods as follows:

---

## [Decision Letter · Decision Letter 1]

3 Feb 2020

Significance of skull osteoporosis to the development of peritumoral brain edema after LINAC-based radiation treatment in patients with intracranial meningioma

PONE-D-19-32359R1

Dear Dr. Han,

We are pleased to inform you that your manuscript has been judged scientifically suitable for publication and will be formally accepted for publication once it complies with all outstanding technical requirements.

With kind regards,

Jonathan H Sherman

Academic Editor

PLOS ONE

Additional Editor Comments (optional):

Reviewers' comments:

Reviewer's Responses to Questions

**Comments to the Author**

1. If the authors have adequately addressed your comments raised in a previous round of review and you feel that this manuscript is now acceptable for publication, you may indicate that here to bypass the “Comments to the Author” section, enter your conflict of interest statement in the “Confidential to Editor” section, and submit your "Accept" recommendation.

Reviewer #1: All comments have been addressed

Reviewer #2: All comments have been addressed

2. Is the manuscript technically sound, and do the data support the conclusions?

Reviewer #1: Yes

Reviewer #2: Yes

3. Has the statistical analysis been performed appropriately and rigorously? 

Reviewer #1: Yes

Reviewer #2: Yes

4. Have the authors made all data underlying the findings in their manuscript fully available?

Reviewer #1: No

Reviewer #2: Yes

5. Is the manuscript presented in an intelligible fashion and written in standard English?

Reviewer #1: (No Response)

Reviewer #2: Yes

6. Review Comments to the Author

Reviewer #1: (No Response)

Reviewer #2: All comments have been addressed and thanks for your quick turn around on the edits. I have also learned something new about cumulative hazard curves, so thank you.

7. PLOS authors have the option to publish the peer review history of their article (what does this mean?). If published, this will include your full peer review and any attached files.

Reviewer #1: No

Reviewer #2: Yes: Yuan James Rao

---

## [Editor Report · Acceptance letter]

5 Feb 2020

PONE-D-19-32359R1 

Significance of skull osteoporosis to the development of peritumoral brain edema after LINAC-based radiation treatment in patients with intracranial meningioma 

Dear Dr. Han:

I am pleased to inform you that your manuscript has been deemed suitable for publication in PLOS ONE. Congratulations! Your manuscript is now with our production department. 

With kind regards,

on behalf of

Dr. Jonathan H Sherman 

Academic Editor

PLOS ONE